# Reinforced Molecular Optimization with Neighborhood-Controlled Grammars

**Chencheng Xu,**[1,2] **Qiao Liu,**[1,3] **Minlie Huang,**[1,2*] **Tao Jiang**[4,1,2*]
[1]BNRIST, Tsinghua University, Beijing 100084, China
[2]Department of Computer Science and Technology, Tsinghua University, Beijing 100084, China
[3]Department of Automation, Tsinghua University, Beijing 100084, China
[4]Department of Computer Science and Engineering, UCR, CA 92521, USA
{xucc18, liu-q16}@mails.tsinghua.edu.cn
aihuang@tsinghua.edu.cn, jiang@cs.ucr.edu

## Abstract

A major challenge in the pharmaceutical industry is to design novel molecules with specific desired properties, especially when the property evaluation is costly. Here, we propose MNCE-RL, a graph convolutional policy network for molecular optimization with molecular neighborhood-controlled embedding grammars through reinforcement learning. We extend the original neighborhood-controlled embedding grammars to make them applicable to molecular graph generation and design an efficient algorithm to infer grammatical production rules from given molecules. The use of grammars guarantees the validity of the generated molecular structures. By transforming molecular graphs to parse trees with the inferred grammars, the molecular structure generation task is modeled as a Markov decision process where a policy gradient strategy is utilized. In a series of experiments, we demonstrate that our approach achieves state-of-the-art performance in a diverse range of molecular optimization tasks and exhibits significant superiority in optimizing molecular properties with a limited number of property evaluations.

## 1   Introduction

Traditional drug discovery relies on the development and exploration by expert chemists and pharmacologists, which is time-consuming due to the large chemical structure space [33]. Effective methods for collecting chemical structures with desired properties will significantly reduce the number of candidates for wet-lab experiments and thus accelerate the development of novel drugs.

Recently, several methods have been proposed to solve the molecular optimization problem within the deep learning framework [16, 36, 17, 19, 30, 10]. The major challenges for molecular optimization mainly lie in generating valid molecular structures and efficiently exploring the vast chemical structure space. Although several methods, including [34, 19, 16, 36], have been proposed to solve the first challenge, they either involve complex network architectures or struggle to optimize properties due to the choices of molecular representations [16, 19]. The second challenge is addressed by Bayesian optimization (BO) [16, 19, 25] and reinforcement learning (RL) [36]. However, few of these methods considered the high cost to evaluate molecular properties in real-world applications [17]. In fact, for most chemical and biological properties, such as antibacterial, anticancer and teratogenicity, there are no known explicit functions to directly interpret a chemical structure as a corresponding numerical property score. Hence, time-consuming wet-lab experiments or simulations are typically required to evaluate these properties of molecules, resulting in a limited number of molecules with validated

---

[*]Minlie Huang and Tao Jiang are the co-corresponding authors.

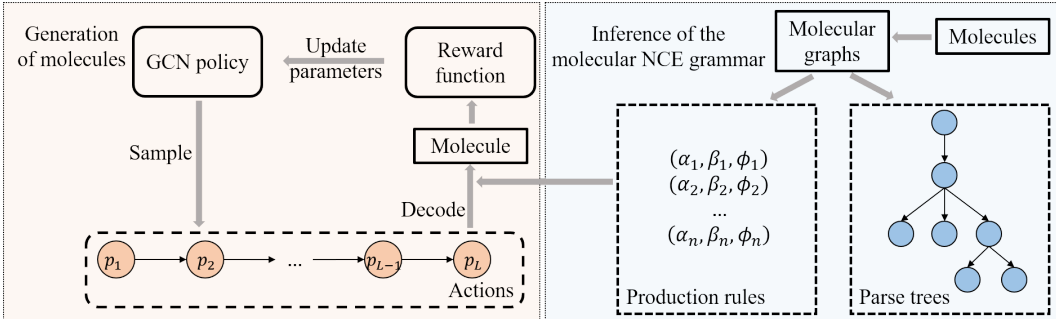

Figure 1: Illustration of our framework. We first infer a molecular NCE grammar by representing molecules as molecular graphs, parsing the graphs using neighborhood-controlled rules, and extracting the production rules. In the generation process, a GCN-based policy network samples a sequence of productions from the action space and obtains rewards from the reward function. The reward function measures the specific property of the molecule decoded from the generated action sequence.

properties. Therefore, generating molecules with desired properties using a small number of property evaluations as well as a small number of molecules with known properties is critical.

To tackle these challenges, we propose MNCE-RL, an RL-based framework using the proposed molecular neighborhood-controlled embedding grammars and a graph convolutional network (GCN). The molecular neighborhood-controlled embedding graph grammars are extended from neighborhood-controlled embedding (NCE) grammars [7, 14], which are a type of sequential context-free graph grammars. As shown in Figure 1, a molecular NCE grammar can be inferred from the input molecular graphs so that each molecule can be represented as a parse tree. In the generation process, an RL agent generates a sequence of production rules, and receives a reward from the environment, which measures the specific property of the generated molecule, that can be used to update the GCN policy network. Our proposed molecular NCE grammars guarantee the chemical validity and the RL agent can efficiently explore the vast chemical structure space.

Our major contributions include 1) a novel molecular NCE grammar and an efficient algorithm to infer production rules from given molecules, where the grammar provides a way to simplify the generation of valid molecules; 2) a novel GCN architecture updating both node and edge features to compute feature vectors for nodes in molecular graphs, where the update of edge features in the GCN makes it possible to capture subtle physical differences between bonds with the same labels and thus lead to better node features for policy decision making; 3) the experimental results show that MNCE-RL significantly outperforms state-of-the-art methods in molecular optimization and has a high potential to be useful in drug discovery.

## 2   Related work

Early methods [30, 10, 4, 11] represent molecules as SMILES strings [34], where the generation of a molecule is modeled as a Markov decision process (MDP) and recurrent neural networks are used to generate the SMILES string. Compared to the graph representation, the SMILES representation is quite brittle as a small change in the string may lead to a completely different molecule, which makes it hard to optimize molecular properties [17]. Winter *et al*. [35] optimize molecular properties in a continuous latent space learned from SMILES strings to overcome the brittleness of the SMILES representation. Li *et al*. [22] first attempt to generate molecules with the graph representation and achieves promising results in generating novel and realistic molecules, but their method cannot guarantee the validity of the generated molecules. To reduce the ratio of invalid molecules, Jin *et al*. [16] (JT-VAE) proposes to represent molecules with junction trees where each node in the tree represents a cluster of atoms and optimize properties in the latent space of the variational autoencoder (VAE) by BO. Although the chemical validity constraints are intrinsically satisfied by predefined connections in clusters, uncertainty in combining the generated clusters limits the model's ability to optimize molecular properties. You *et al*. [36] (GCPN) try to generate molecular graphs by iteratively adding atoms and edges using a graph convolutional policy network and guarantee the chemical validity by the imposition of certain chemical constraints on generated structures. Due

to its complex model architecture, GCPN requires a large number of iterations in training, which limits its applications in situations when property evaluation is costly. Kajino [17] (MHG-VAE) is the first to apply graph grammars to the molecular optimization problem. With a simple VAE architecture, MHG-VAE shows superiority in molecular optimization with a limited number of property evaluations. However, the performance of MHG-VAE is still far from being satisfactory perhaps due to the choice of the grammars and indirect optimization in a latent space.

# 3 Methods

As mentioned in [17], molecular optimization can be formulated as follows:

$$m^* = \arg\max_{m \in \mathcal{M}} f(m), \tag{1}$$

where $\mathcal{M}$ is the set of all valid chemical molecules and $f$ is an evaluation function, which measures some specific property score of molecule $m$. We represent a molecule as a graph $H = (V, E, \sigma, \psi)$ by modeling atoms as nodes and bonds as edges, where $V$ is a finite set of nodes, $E$ is a finite set of edges, $\sigma : V \to \Sigma$ is a node-labeling function, which projects $V$ to the node label set $\Sigma$, and similarly $\psi : E \to \Psi$ is an edge-labeling function that projects $E$ to the edge label set $\Psi$. Following [17], we use the Kekulé structure of molecules and include the chirality tag in node labels.

Using the proposed molecular NCE grammars, the generation of a novel molecule is interpreted as the generation of a parse tree, where each node in the tree represents a production rule. Furthermore, by traversing the parse trees in preorder, the molecular optimization problem is interpreted as the generation of an optimal production sequence, *i.e.*

$$Prod^* = \arg\max_{Prod \in \mathcal{P}} f \circ Dec_{\mathcal{P}}(Prod), \tag{2}$$

where $\mathcal{P}$ is the set of all valid sequences of production rules and $Dec_{\mathcal{P}} : \mathcal{P} \to \mathcal{M}$ is the decoding function that transforms a production sequence into a molecule. The problem can be cast as an MDP and solved in the RL framework, where a GCN is used for node feature aggregation. Given an intermediate production sequence $Prod_t$ generated at time step $t$, due to the constraints of the molecular NCE grammar, the next production rule can only be selected from a subset of the production rules. We denote a production rule to be legal for $Prod_t$ if it satisfying the grammatical constraints.

## 3.1 Problem formulation as reinforcement learning

As aforementioned, the generation of sequences of production rules can be formulated as a sequential decision problem. Hence, we present the design of state representation, action space, and reward function as follows.

**State.** We denote the state $s_t$ at time step $t$ as the intermediate sequence $Prod_t = p_1 p_2 ... p_{t-1}$, from which a graph $H_t$ can be decoded and the non-terminal node $v_t$ to be rewritten at time step $t + 1$ is determined. Note that at the first step, $Prod_1$ is an empty sequence and $H_1$ has only one node $v_1$ with the starting symbol.

**Action.** The action space is a set of the legal production rules for $Prod_t$. In time step $t$, the policy $\pi_\theta(a_t|s_t)$ samples a production rule from the action space, where

$$\pi_\theta(a_t|s_t) = softmax(\mathbf{F}_{\theta'}(H_t)_{v_t}\mathbf{W} + \mathbf{b}), \tag{3}$$

in which $\mathbf{F}$ is a GCN described in section 3.4, $\theta'$ is the parameter set of $\mathbf{F}$, and $\theta = \{\mathbf{W}, \mathbf{b}\} \cup \theta'$. $\mathbf{F}_{\theta'}(H_t)$ is the computed node feature matrix of $H_t$ and $\mathbf{F}_{\theta'}(H_t)_{v_t}$ is the row corresponding to the node $v_t$. The intermediate molecular graph is updated with the sampled production rule.

**Reward.** As the generation process may take too many steps to converge, we set a threshold $T_{max}$ and force the generation process to stop when the number of steps exceeds $T_{max}$. Assume that the length of the generated sequence is $T - 1$. At time step $t < T$, a small constant reward $r_\epsilon$ is assigned and at time step $T$, if there is no non-terminal node in $H_T$, a task-specific reward function assigns a reward based on $f \circ Dec_{\mathcal{P}}(H_T)$. Otherwise, a constant non-positive reward $r_{incomp}$ is assigned.

## 3.2 Definition of molecular NCE grammars

An NCE graph grammar proposed by Janssens et al. [14] is a system $G = (\Sigma, \Delta_\Sigma, P)$, where $\Sigma$ is the set of node labels, and $\Delta_\Sigma \subset \Sigma$ is the terminal alphabet and $P$ is the set of production rules. A

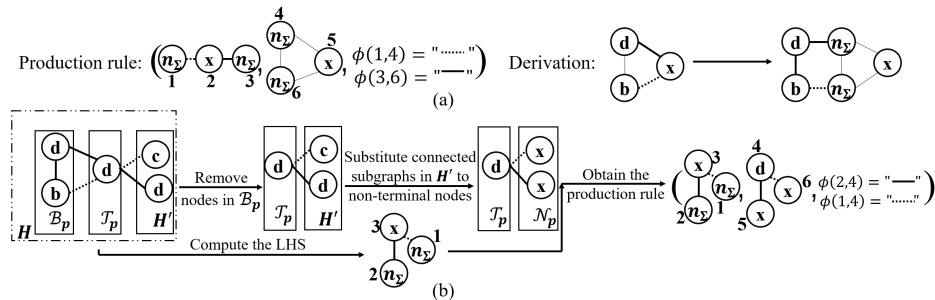

Figure 2: (a) An example production rule and a derivation step. Here, $x$ and $n_\Sigma$ are non-terminal labels. The production rule is in the form of $(\alpha, \beta, \phi)$. In a derivation step, applying a production rule $p$ will replace a non-terminal node $v_t$ (with label $x$) in the intermediate graph $H_t$ with the RHS ($\beta$) of $p$, and the edges between neighbors of $v_t$ and nodes in $V_\beta$ is determined by the embedding function $\phi$. (b) Extraction of a production rule. $\mathcal{B}_p$, $\mathcal{T}_p$ and $\mathcal{N}_p$ are vertex sets. $H'$ is a node-induced subgraph of $H$. The LHS is obtained by representing the nodes in $\mathcal{T}_p$ as a non-terminal node, removing the edges between the nodes in $\mathcal{B}_p$ and labeling the nodes in $\mathcal{B}_p$ as $n_\Sigma$. The RHS is obtained by removing the nodes in $\mathcal{B}_p$ from $H$ and replacing the connected subgraphs in $H'$ by non-terminal nodes.

production rule is in the form of $p = (\alpha, \beta, \phi)$, where $\alpha$, $\beta$ are connected graphs. $\alpha$ is called the left-hand side (LHS) of $p$, $\beta$ is called the right-hand side (RHS), and $\phi : V_\alpha \times V_\beta \times \Sigma \to \{0, 1\}$ is the embedding function. Directly applying NCE grammars to molecular graphs suffers from the following issues: 1) A molecular graph is both node-labeled and edge-labeled, while the NCE grammars are defined only on node-labeled graphs. 2) The connections between the neighbors of $V_\alpha$ and nodes in $V_\beta$ are not specified, which may cause valency invalidity in a molecular graph. 3) The number of production rules may explode, decreasing the generalization ability of the grammars. To extend NCE grammars to molecular graphs, we define molecular NCE grammars as follows.

**Definition 1** *A molecular NCE grammar is a system $G = (\Sigma, \Psi, \Delta_\Sigma, \Delta_\Psi, P)$, where $\Sigma$ is the set of node labels, $\Psi$ the set of edge labels, $\Delta_\Sigma = \Sigma \setminus \{x, n_\Sigma, s\}$ the terminal alphabet of nodes, $\Delta_\Psi = \Psi \setminus \{n_\Psi\}$ the terminal alphabet of edges, $s$ the starting symbol, and $n_\Sigma$ and $n_\Psi$ the empty labels for nodes and edges, respectively. Finally, $P$ is the set of production rules. A production rule is in the form of $p = (\alpha, \beta, \phi)$ where:*

- *$\alpha = (V_\alpha, E_\alpha, \sigma_\alpha, \psi_\alpha, L_\alpha)$ and $\beta = (V_\beta, E_\beta, \sigma_\beta, \psi_\beta, L_\beta)$ are ordered connected graphs, where $L_*$ defines a unique order for edges incident to each vertex in the graph*

- *$V_\alpha = \{X_p\} \cup \mathcal{B}_p$, $E_\alpha = \{X_p\} \times \mathcal{B}_p$, where $X_p$ is a non-terminal node with $\sigma_\alpha(X_p) = x$ and $\mathcal{B}_p$ is a set of nodes with $\forall v \in \mathcal{B}_p$, $\sigma_\alpha(v) = n_\Sigma$*

- *$V_\beta = \mathcal{T}_p \cup \mathcal{N}_p$, $E_\beta \subset (\mathcal{T}_p \times \mathcal{T}_p) \cup (\mathcal{T}_p \times \mathcal{N}_p)$, where $\mathcal{T}_p$ and $\mathcal{N}_p$ are sets of nodes with $\forall v \in \mathcal{N}_p, \sigma_\beta(v) = x$*

  - *if $\|\mathcal{T}_p\| > 1$, then $\forall u \in \mathcal{T}_p, \sigma_\beta(u) = n_\Sigma$, $\forall e \in E_\beta, \psi_\beta(e) = n_\Psi$, $p$ is a complex production rule*
  - *if $\|\mathcal{T}_p\| = 1$, then $\forall u \in \mathcal{T}_p, \sigma_\beta(u) \in \Delta_\Sigma$, $\forall e \in E_\beta, \psi_\beta(e) \in \Delta_\Psi$, $p$ is a simple production rule*

- *$\phi : \mathcal{B}_p \times \mathcal{T}_p \to \Psi \cup \{0\}$ is the embedding function*

The first two issues mentioned above are addressed by specifying $\psi_\alpha$, $\psi_\beta$ and $\phi$. To alleviate the third issue, we introduce the empty labels that can be matched more arbitrarily, $n_\Sigma$ and $n_\Psi$, in a more general way. The labels of nodes in $\mathcal{B}_p$ are replaced by $n_\Sigma$ and for complex production rules, only the skeletons of $\beta$ are kept. Production rules predefine the edges incident to each vertex and thus the valency validity can be guaranteed intrinsically. To specify the action space at each step, we define legal production rules as follows.

**Definition 2** *Let $T_t$ be an intermediate tree. If $T_t$ is an empty tree, the legal production rules for $T_t$ is the set of starting production rules. If $T_t$ is not empty and we need to sample a child*

*production rule for the parent $p_{parent}$ that already has a set of child production rules $P_{sibling}$, then an intermediate graph $H_t$ with a non-terminal node $v_t$ to be rewritten at in the next time step can be decoded from $T_t$. Suppose that the direct neighbors of $v_t$ are $\{v_{n_1}, v_{n_2}, ..., v_{n_k}\}$ and $L_t$ sorts the edge set $\{(v_t, v_{n_1}), (v_t, v_{n_2}), ..., (v_t, v_{n_k})\}$ in the order in which $v_{n_i}$ are generated, we say that a production rule $p$ matches the context of $v_t$ if and only if the edge-induced subgraph of $H_t$ specified by $\{(v_t, v_{n_1}), (v_t, v_{n_2}), ..., (v_t, v_{n_k})\}$ and ordered by $L_t$ is isomorphic to the LHS of $p$ [15]. Then*

- *if $p_{parent}$ is complex and $v_t \in \mathcal{T}_{p_{parent}}$, any production rule having a positive empirical probability $P(p|p_{parent}, P_{sibling})$ and matching the context of $v_t$ is legal for $T_t$*

- *otherwise, any production rule matching the context of $v_t$ is legal for $T_t$*

An example production rule and a derivation step are shown in Figure 2. Applying a production rule $p$ to an intermediate graph $H_t$ to rewrite a non-terminal node $v_t$ will replace $v_t$ with the RHS of $p$, and the edges between the direct neighbors of $v_t$ and nodes in the RHS are specified by the embedding function. A formal notion of a derivation step is defined as follows.

**Definition 3** *Let $T_t$ be an intermediate parse tree and a production rule $p = (\alpha, \beta, \phi)$ is legal for $T_t$. An intermediate graph $H_t$ and a non-terminal node $v_t$ can be decoded from $T_t$. A derivation step of applying $p$ to $H_t$ will generate a novel graph $H_{t+1}$ by rewriting the node $v_t$, where*

- $V_{H_{t+1}} = V_{H_t} \cup V_\beta \setminus \{v_t\}$

- $E_{H_{t+1}} = \{(u,v)|u,v \in V_{H_t} \setminus \{v_t\} \text{ and } (u,v) \in E_{H_t}\} \cup E_\beta \cup \{(u,v)|\phi(u,v) \in \Psi\}$

*For a node $u \in V_{H_{t+1}}$ and an edge $(u,v) \in E_{H_{t+1}}$, the labeling functions are:*

$$\begin{cases} \sigma_{H_{t+1}}(u) = \sigma_{H_t}(u), \ u \in V_{H_t} \\ \sigma_{H_{t+1}}(u) = \sigma_\beta(u), \ u \in V_\beta \end{cases} , \begin{cases} \psi_{H_{t+1}}((u,v)) = \psi_{H_t}((u,v)), \ (u,v) \in E_{H_t} \\ \psi_{H_{t+1}}((u,v)) = \psi_\beta((u,v)), \ (u,v) \in E_\beta \\ \psi_{H_{t+1}}((u,v)) = \phi(u,v), \ u \in V_{H_t}, v \in V_\beta, \phi(u,v) \in \Psi \end{cases}$$

With this definition, by learning production rules from known molecules, any molecule sampled from the inferred grammar is chemically valid. A comparison of our proposed grammars and the MHGs [17] is shown in Appendix B.

### 3.3 Inference of the molecular NCE grammars

The algorithm to parse molecular graphs and infer the production rules is shown in Appendix B. We sort the nodes of $H$ in the depth-first (DF) order, and for a node $v$ with first-hop neighbors $\{v_{n_1}, v_{n_2}, ..., v_{n_k}\}$, the edges $\{(v, v_{n_1}), (v, v_{n_2}), ..., (v, v_{n_k})\}$ are sorted to be consistent with the order of $v_{n_i}$. The graph is parsed in the DF order and the LHS and RHS extracted from $H$ inherit the edge orders. For a simple production rule (Figure 2), the LHS is simply obtained by representing nodes in $\mathcal{T}_p$ as a non-terminal node, removing the edges between the nodes in $\mathcal{B}_p$ and labeling nodes in the $\mathcal{B}_p$ as $n_\Sigma$. The embedding function $\phi$ is obtained by recording the edges between the nodes in $\mathcal{T}_p$ and $\mathcal{B}_p$. Denoting the node-induced subgraph of $H$ specified by $V_H \setminus (\mathcal{B}_p \cup \mathcal{T}_p)$ as $H'$, the RHS is obtained by removing the nodes in $\mathcal{B}_p$ from $H$ and representing each connected subgraph of $H'$ with a non-terminal node. For the complex production rules (Appendix A), the first steps are also computing the LHS, recording the embedding function, removing nodes in $\mathcal{B}_p$, and substitute connected subgraphs in $H'$ into non-terminal nodes. In the final step, as discussed in the prior section, to reduce the number of production rules, we only keep the skeleton of the RHS, and the labels of all nodes in $\mathcal{T}_p$ and the labels of all edges in the RHS are replaced by $n_\Sigma$ and $n_\Psi$. To maintain the information, we introduce an extra production rule for each node in $\mathcal{T}_p$. Examples to parse a molecular graph and to sample a molecule from a grammar is shown in Appendix A.

### 3.4 Graph convolutional network for node feature aggregation

Graph convolutional networks (GCNs) [9, 12, 23, 21, 8, 18] have been widely applied in graph information aggregation. We represent both nodes and edges with feature vectors. In the forward pass, the GCN updates both the node features and the edge features and outputs the computed features for

all nodes in the last layer. Assuming that the feature size of the edges is $S_E$, the node features are updated by

$$\mathbf{V}^{(l+1)} = AGG(\{Tanh(\mathbf{E}_{(i)}^{(l)}\mathbf{V}^{(l)}\mathbf{W}_{(i)}^{(l+1)} + \mathbf{b}_{(i)}^{(l+1)}) + \mathbf{V}^{(l)} | i \in (1, ..., S_E)\}), \qquad (4)$$

where $AGG$ is the aggregation function, $\mathbf{V}^{(l)}$ is the node feature matrix in the $l$th layer, $\mathbf{E}_{(i)}^{(l)}$ is the $i$th feature matrix of edges, and $\mathbf{W}_*^*$ and $\mathbf{b}_*^*$ are parameters of the network. The edge features are updated in two steps. At the first step, we calculate a vector $\mathbf{e}_{ij}$, which encodes the relationship between the $i$-th node and the $j$-th node using the following formula

$$\mathbf{e}_{ij}^{(l+1)} = ReLU(Concat(\mathbf{V}_i^{(l+1)}, \mathbf{V}_j^{(l+1)})\mathbf{W}_e^{(l+1)} + \mathbf{b}_e^{(l+1)}), \qquad (5)$$

where $\mathbf{V}_i^{(l+1)}$ is the feature vector of the node $i$ in the $(l+1)$th layer. Then, the feature vector $\mathbf{E}_{ij}$ of the edge between the node $i$ and the node $j$ is updated by

$$\mathbf{E}_{ij}^{(l+1)} = ReLU(Concat(\mathbf{e}_{ij}^{(l+1)}, \mathbf{E}_{ij}^{(l)})\mathbf{W}_E^{(l+1)} + \mathbf{b}_E^{(l+1)}). \qquad (6)$$

### 3.5 Model training

To generate molecules with desired properties, the widely used RL technique, Proximal Policy Optimization [29] (PPO), is adopted to train the model. The objective function of PPO is

$$L^{CLIP}(\theta) = \hat{E}_t \left[ \min(r_t(\theta)\hat{A}_t, clip(r_t(\theta), 1 - \epsilon, 1 + \epsilon)\hat{A}_t) \right], \qquad (7)$$

where $\epsilon$ is a hyperparameter, $\theta$ is the policy parameter, $\hat{E}_t$ denotes the empirical expectation over timesteps, and $r_t$ is the ratio of the probability under the new and old policies, i.e.

$$r_t = \frac{\pi_\theta(a_t|s_t)}{\pi_{\theta_{old}}(a_t|s_t)}, \qquad (8)$$

where $\theta_{old}$ is the parameter set of the old policy. $\hat{A}_t$ is the estimated advantage [28] at time step $t$. We compute the actor critic $C_\omega(\cdot)$ in $\hat{A}_t$ as

$$C_\omega(H_t) = Avg(\mathbf{F}_{\omega'}(H_t))\mathbf{W_C} + \mathbf{b_C}, \qquad (9)$$

where $\mathbf{F}$ is a GCN with the parameter set $\omega'$, $\omega$ is the parameter set of the actor critic and $\omega = \{\mathbf{W_C}, \mathbf{b_C}\} \cup \omega'$. The $Avg$ function computes the average over the node features. To encourage the model to generate graphs with high diversity, an entropy loss [24] is also added to the loss function, and to accelerate convergence, we take all the ground truth molecules as expert trajectories and pre-train the model with these trajectories. Details of model training and optimizations of hyperparameters are shown in Appendix F.

## 4 Experiments

### 4.1 Datasets

The ZINC250k molecule dataset [13], GuacaMol package [3] and 2,337 drug molecules from [31] are used in our experiments. The ZINC250k dataset contains 250,000 drug-like molecules whose maximum atom number is 38. The work in [31] provides 2,337 drug molecules and their inhibition effects to *E.coli* collected from wet-lab experiments. With a threshold of 0.2, 120 of the 2,337 molecules that have a strong *E.coli* growth inhibition are defined as the positive set and the remaining molecules are considered as the negative set. GuacaMol is a comprehensive benchmark package for molecular optimization that provides more than one million molecules and covers not only single-objectives but also constrained and multi-objective optimization tasks. The validity of generated molecules is checked by RDKit [20]. The statistics of the inferred molecular NCE grammars are provided in Appendix C[2].

## 4.2 Molecular optimization results

To demonstrate the ability of MNCE-RL in molecular optimization in different application scenarios, we designed a series of experiments and compared MNCE-RL with the current state-of-the-art methods. Detailed experiment settings of the baseline models [35, 17, 16, 36] are provided in Appendix D.

**Property optimization with unlimited evaluations and an ablation study.** In this experiment, we assume that the cost of property evaluation is negligible and the number of times to query the molecule properties is unlimited. Penalized logP score and QED score are used to evaluate the performance of models. Here, LogP is an estimation of the octanol-water partition coefficient and penalized logP also accounts for ring size and synthetic accessibility [6]. QED [2] is a computational score for measuring the drug-likeness of a molecule. To measure the performance of each method, we report the top 3 property scores, the 50th best score, and the average score of the top 50 molecules. The task-specific reward function we used in our approach is a linear projection of the computed penalized logP or QED score. The results are shown in Table 1 and Appendix G. To investigate the specific contributions of our proposed grammars and the GCN structure in this experiment, we build a model using the classical GCN [9, 36] without edge feature updating (MNCE-RL$_{OEU}$). As shown in the tables, MNCE-RL$_{OEU}$ achieves the state-of-the-art performance in optimizing both penalized logP and QED and significantly outperforms GCPN, indicating the effectiveness of our grammars. Moreover, compared with the MHGs, our proposed grammars achieve a higher coverage rate (Appendix C), and thus can represent more molecular structures and explore the chemical space more effectively. The utility of the edge feature updating mechanism is also confirmed by the fact that MNCE-RL outperforms MNCE-RL$_{OEU}$ significantly in optimizing penalized logP.

Table 1: Results on property optimizations with unlimited property evalutions

| Method | Penalized logP | | | | | | QED | | | | | |
|---|---|---|---|---|---|---|---|---|---|---|---|---|
| | $1^{st}$ | $2^{nd}$ | $3^{rd}$ | $50^{th}$ | Top 50 Avg. | Validity | $1^{st}$ | $2^{nd}$ | $3^{rd}$ | $50^{th}$ | Top 50 Avg. | Validity |
| JT-VAE | 5.30 | 4.93 | 4.49 | 3.50 | 3.93 | **100%** | 0.942 | 0.934 | 0.930 | 0.896 | 0.912 | **100%** |
| GCPN | 7.98 | 7.85 | 7.80 | - | - | **100%** | **0.948** | 0.947 | 0.946 | - | - | **100%** |
| MHG-VAE | 5.56 | 5.40 | 5.34 | 4.12 | 4.49 | **100%** | 0.947 | 0.946 | 0.944 | 0.920 | 0.929 | **100%** |
| MSO | 14.44 | 14.20 | 13.95 | 13.49 | 13.67 | - | **0.948** | **0.948** | **0.948** | **0.948** | **0.948** | - |
| MNCE-RL$_{OEU}$ | 14.49 | 14.44 | 14.36 | 14.13 | 14.16 | **100%** | **0.948** | **0.948** | **0.948** | **0.948** | **0.948** | **100%** |
| MNCE-RL | **18.33** | **18.18** | **18.16** | **17.52** | **17.76** | **100%** | **0.948** | **0.948** | **0.948** | **0.948** | **0.948** | **100%** |

**Constrained property optimization.** This task aims at generating molecules with an improved penalized logP score while keeping structures similar to a given target molecule. Different from previous methods, such as GCPN, that can generate novel molecules starting from a given molecule, we first train our model to maximize the log-likelihood of the target molecule and then optimize the penalized logP. The task-specific reward assigns a small constant score if the similarity drops below the threshold and assigns a linear projection of the penalized logP score if the similarity is larger than the threshold. The results are shown in Table 2 and Appendix G, where the $\delta$ is the threshold of the similarity score. MNCE-RL is capable to optimize all the molecules with success rates of 100% on both thresholds and for each threshold, MNCE-RL achieves significantly higher improvements in penalized logP than all baseline models. Although the average similarity scores of the molecules generated by MNCE-RL are slightly lower than those generated by the baseline models, the improvements in penalized logP achieved by MNCE-RL with similarity threshold 0.6 is significantly higher than baseline models with threshold 0.4, exhibiting the superiority of MNCE-RL.

**Comprehensive evaluations with GuacaMol.** These experiments comprehensively measure a model's ability in optimizing properties with unlimited evaluations. The results are shown in Table 3, where BNGM represents the best results of the naive baselines provided in the manuscript of GuacaMol [3]. The performance of MNCE-RL exceeds the baselines on all benchmarks. In particular, our method significantly outperforms the baselines in multi-objective optimization tasks, showing the superiority of MNCE-RL in complex scenarios.

Table 2: Results on constrained property optimizations

| Method | $\delta = 0.4$ | | | $\delta = 0.6$ | | |
|---|---|---|---|---|---|---|
| | Improvement | Similarity | Success | Improvement | Similarity | Success |
| JT-VAE | $0.84 \pm 1.45$ | $0.51 \pm 0.10$ | 83.6% | $0.21 \pm 0.71$ | $0.69 \pm 0.06$ | 46.4% |
| GCPN | $2.49 \pm 1.30$ | $0.47 \pm 0.08$ | **100%** | $0.79 \pm 0.63$ | $0.68 \pm 0.08$ | **100%** |
| MHG-VAE | $1.00 \pm 1.87$ | **$0.52 \pm 0.11$** | 43.5% | $0.61 \pm 1.20$ | **$0.70 \pm 0.06$** | 17.0% |
| MNCE-RL | **$5.29 \pm 1.58$** | $0.45 \pm 0.05$ | **100%** | **$3.87 \pm 1.43$** | $0.64 \pm 0.04$ | **100%** |

Table 3: Results on the benchmarks provided by GuacaMol.

| Benchmark | Methods | | | Benchmark | Methods | | |
|---|---|---|---|---|---|---|---|
| | BNGM | MSO | MNCE-RL | | BNGM | MSO | MNCE-RL |
| Celecoxib rediscovery | **1.0** | **1.0** | **1.0** | Osimertinib MPO | 0.953 | 0.966 | **1.0** |
| Troglitazone rediscovery | **1.0** | **1.0** | **1.0** | Fexofenadine MPO | 0.998 | **1.0** | **1.0** |
| Thiothixene rediscovery | **1.0** | **1.0** | **1.0** | Ranolazine MPO | 0.920 | 0.931 | **0.990** |
| Aripiprazole similarity | **1.0** | **1.0** | **1.0** | Perindopril MPO | 0.808 | 0.834 | **0.882** |
| Albuterol similarity | **1.0** | **1.0** | **1.0** | Amlodipine MPO | 0.894 | 0.900 | **0.920** |
| Mestranol similarity | **1.0** | **1.0** | **1.0** | Sitagliptin MPO | 0.891 | 0.868 | **0.904** |
| C11H24 | 0.993 | 0.997 | **1.0** | Zaleplon MPO | 0.754 | 0.764 | **0.781** |
| C9H10N2O2PF2Cl | 0.982 | **1.0** | **1.0** | Valsartan SMARTS | 0.990 | 0.994 | **1.0** |
| Median molecules 1 | 0.438 | 0.437 | **0.455** | Scaffold Hop | **1.0** | **1.0** | **1.0** |
| Median molecules 2 | 0.432 | 0.395 | **0.457** | Deco Hop | **1.0** | **1.0** | **1.0** |

**Property range targeting.** This experiment measures the model's ability to generate diverse molecules with some specific property in a predefined range [36], where the diversity is defined as the average pairwise Tanimoto distance between the Morgan fingerprints of the generated molecules [26]. Penalized logP and molecular weight (MW) are considered in this task where the predefined ranges are the same as those used in [36]. The task-specific reward in our approach is inversely proportional to the distance between the property score of a generated molecule and the center of the predefined range. The results are shown in Table 4. Our model achieves over 90% success rates in all the four tasks with high diversities [36] and an over 99% success rate in targeting the range $500 \leq MW \leq 550$, which significantly outperforms state-of-the-art methods.

Table 4: Results on property range targeting

| Method | $-2.5 \leq logP \leq -2$ | | $5 \leq logP \leq 5.5$ | | $150 \leq MW \leq 200$ | | $500 \leq MW \leq 550$ | |
|---|---|---|---|---|---|---|---|---|
| | Success | Diversity | Success | Diversity | Success | Diversity | Success | Diversity |
| JT-VAE | 11.3% | **0.846** | 7.6% | **0.907** | 0.7% | 0.824 | 16.0% | 0.898 |
| GCPN | 85.5% | 0.392 | 54.7% | 0.855 | 76.1% | 0.921 | 74.1% | **0.920** |
| MNCE-RL | **98.3%** | 0.836 | **98.0%** | 0.842 | **91.8%** | **0.928** | **99.6%** | 0.870 |

**Property optimization with limited property evaluations.** This task measures a model's ability to optimize molecules when the property evaluation is expensive. As done in [17], we limit the number of molecule property queries to 500. We repeat MNCE-RL ten times and take the first 500 molecules generated as the output each time to obtain a total of 5k molecules. The task-specified reward is the same as in property optimization with unlimited property evaluations. The top 3 property scores, the 50th best score and the average score of the top 50 molecules are recorded. The results are shown in Table 5 and Appendix G. Our model significantly outperforms all baseline methods. Interestingly, even with limited property evaluations, our method still performs better than JT-VAE and MHG-VAE with unlimited evaluations. Moreover, the top 50 scored molecules generated by MNCE-RL has a higher average penalized logP score than the top-scored molecule generated by all baselines, which demonstrates the superiority of MNCE-RL in situations when it is expensive to evaluate molecular properties.

Table 5: Results on property optimization with limited property evaluations

| Method | Penalized logP | | | | | Validity |
| | $1^{st}$ | $2^{nd}$ | $3^{rd}$ | $50^{th}$ | Top 50 Avg. | |
|---|---|---|---|---|---|---|
| JT-VAE | 1.69 | 1.68 | 1.60 | -9.93 | -1.33 | **100%** |
| GCPN | 2.77 | 2.73 | 2.34 | 0.91 | 1.36 | **100%** |
| MHG-VAE | 5.24 | 5.06 | 4.91 | 4.25 | 4.53 | **100%** |
| MSO | 2.96 | 2.91 | 2.75 | 2.49 | 2.54 | **100%** |
| MNCE-RL | **9.88** | **9.82** | **9.75** | **7.28** | **8.31** | **100%** |

**Generation of novel molecules with antibacterial property.** This experiment shows MNCE-RL's ability to assist drug discovery in a real-world application scenario when the number of experimentally validated molecules is limited and there is no known evaluation function. We first train a classifier on the 2,337 molecules from [31] to distinguish positive and negative samples and use the classifier as a pseudo evaluation function. Then, we extract production rules from these molecules. The problem is modeled as a property optimization where we try to find molecules that receive high scores from the classifier. As the classifier is severely overfitted, when training the generation model, we assume that the generated novel molecules are negative and use these "negative samples" to update the classifier to reduce bias. After training, the kinase inhibitor scores, the protease inhibitor scores, and the enzyme inhibitor scores [27, 32, 5, 1] (see Appendix E for details) of the top 10 molecules with the highest scores assigned by the classifier are reported. The results are shown in Appendix G and Table 6. Ten of the ten molecules are bioactive (with scores larger than 0.2; see Table 6) with at least one inhibitor score, and six of them are highly bioactive (with scores larger than 0.5), which illustrates the ability of MNCE-RL to generate antibacterial candidate molecules with only limited labeled samples.

Table 6: Properties of six molecules having high inhibitor scores.

| Molecule | Computed properties | | |
| | Kinase inhibitor (KI) | Protease inhibitor (PI) | Enzyme inhibitor (EI) |
|---|---|---|---|
| $M_1$ | -0.38 | 0.56 | 0.25 |
| $M_2$ | -0.20 | 0.54 | 0.23 |
| $M_3$ | -0.34 | 0.55 | 0.15 |
| $M_4$ | -0.24 | 0.63 | 0.09 |
| $M_5$ | -0.16 | 0.66 | 0.16 |
| $M_6$ | -0.24 | 0.66 | 0.30 |

# 5   Conclusion and future work

In this paper, we propose a new method MNCE-RL based on the novel molecular NCE grammars to solve the molecular optimization problem in the RL framework. MNCE-RL achieves the state-of-the-art performance in a series of systematic experiments. In a real-world application, when the molecules with known properties are limited and no numerical evaluation function is known, our method still exhibits high potential to generate molecules with desired properties, showing its great potential utility in drug discovery. Although our proposed grammar guarantees the valency validity of the generated structures, it struggles to capture high-level chemical properties such as bond orders. We leave it to future work.

## Broader impact

Finding effective medicines for diseases has always been a challenge in the pharmaceutical industry, especially when precision medicine has attracted more and more attention in recent years. Our approach provides an efficient way to generate molecules with specific properties, which will help

reduce the workload of pharmacists, accelerate the development of novel drugs, and decrease the cost of drug design. On the other hand, although the molecules generated by our method possess desirable biological or chemical properties, their safety and effectiveness on patients still need to be validated in the normal clinical trial processes.

## Acknowledgments and Disclosure of Funding

This work has been supported in part by the National Natural Science Foundation of China grant 61772197, the National Key Research and Development Program of China grant 2018YFC0910404 and the Guoqiang Institute of Tsinghua University with grant no. 2019GQG1.

## Footnotes

[2]Link to code and datasets: https://github.com/Zoesgithub/MNCE-RL

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
