[Supplementary Material]

# Appendix A  Supplementary figures

Figure A.1: Extraction of complex production rules. The LHS is computed by representing the nodes in $\mathcal{T}_p$ as a non-terminal node, removing the edges between nodes in $\mathcal{B}_p$ and labeling the nodes in $\mathcal{B}_p$ as $n_\Sigma$. The computation of RHS is removing the nodes in $\mathcal{B}_p$ and turning the connected subgraphs in $H'$ to non-terminal nodes. To reduce the number of production rules, only the skeleton of the RHS is kept and a production rule for each node in $\mathcal{T}_p$ is introduced to maintain the information.

Figure A.2: An example of transforming a molecule into a parse tree and inferring molecular NCE grammar production rules.

Figure A.3: An example of sampling a molecule from a molecular NCE grammar. The production rules are shown in Figure A.2.

## Appendix B  Supplementary information of the proposed grammars

The algorithm to infer production rules of the molecular NCE grammar and parse molecular graphs into parse trees is shown in Algorithm 1, where $v_T$ is a node of the parse tree $T$ and $Neigh(v)_H$ is the set of first-hop neighbors of a node $v$ in the graph $H$. For a molecular graph $H$ with $\|V_H\|$ nodes, the time complexity of Algorithm 1 is $O(\|V_H\|^2)$.

Compared with MHG, our proposed grammars have better generalization ability. MHG is an extension of hyperedge replacement grammar, which is based on the clique tree decomposition of graphs. In molecular hypergraphs, the clique tree decomposition might introduce a large number of rare substructures and cause a low coverage rate. For instance, in the MHG inferred from the ZINC250k dataset, 1,424/2,031 are starting rules and 2/3 of the starting rules are used by less than ten molecules. At the same time, 16/5,000 molecules in the testing set cannot be covered by these inferred production rules. In comparison, our grammar is based on neighboring relationships. In molecular graphs, the degree and neighbors of each node are limited by chemical rules, thus the substructure involved in our grammar is relatively simpler and in smaller fragments, which leads to fewer production rules and a higher coverage rate (see Appendix C).

In the generation process, to be consistent with the inference process, the non-terminals with labels of $n_\Sigma$ have higher priority than non-terminals with labels of $x$, and the non-terminals that are generated later have higher priority. The non-terminal with the highest priority in the intermediate graph is rewritten each time.

---

**Algorithm 1:** Inference of molecular NCE grammar production rules

---

**Input:** $H$, $P$, $\mathcal{B}_p$, $\mathcal{T}_p$, $v_T$
**Output:** $T$, $P$
**Function** `ParseMolecularGraph`($H$, $P$, $\mathcal{B}_p$, $\mathcal{T}_p$, $v_T$):

  **if** $\mathcal{B}_p$ *is empty* **then**
      Initialize tree $T$;
      Initialize $v_T$ as the root of $T$;
      Add the initial node to $\mathcal{B}_p$;
      Arbitrarily select a node from $H$ and add it to $\mathcal{T}_p$;
  **end**
  Compute $LHS$;
  Record the embedding function $\phi$;
  Denote $H'$ as a node-induced subgraph of $H$ where $V_{H'} = V_H \setminus (\mathcal{B}_p \cup \mathcal{T}_p)$;
  Remove nodes in $\mathcal{B}_p$ from $H$;
  Represent connected subgraphs in $H'$ by non-terminal nodes;
  Obtain the $RHS$;
  **if** $\|\mathcal{T}_p > 1\|$ **then**
      **for** $v$ *in* $\mathcal{T}_p$ **do**
          Add a child node $v_c$ to $v_T$;
          Extract a production rule $p_v$ for $v$;
          Label $v_c$ as $p_v$;
          Add $p_v$ to $P$;
      **end**
      $RHS \longleftarrow$ The skeleton of $RHS$;
  **end**
  $p \longleftarrow (LHS, RHS, \phi)$;
  Label $v_T$ as $p$;
  Add $p$ to $P$;
  $\mathcal{T}^{(descent)} \longleftarrow \cup_{v \in \mathcal{T}_p} Neigh(v)_H$;
  **for** *connected subgraph $h$ in $H'$* **do**
      Add a child node $v_c$ to $v_T$;
      $\mathcal{B}^{(h)} \longleftarrow (\cup_{v \in V_h} Neigh(v)_H) \setminus V_h$;
      $\mathcal{B}_p^{(h)} \longleftarrow \mathcal{T}_p \cap \mathcal{B}^{(h)}$;
      $\mathcal{T}_p^{(h)} \longleftarrow \mathcal{T}^{(descent)} \cap V_h$;
      Denote $H^{(h)}$ as an induced subgraph of $H$, where $V_{H^{(h)}} = V_h \cup \mathcal{B}_p^{(h)}$;
      `ParseMolecularGraph`($H^{(h)}$, $P$, $\mathcal{B}_p^{(h)}$, $\mathcal{T}_p^{(h)}$, $v_c$);
  **end**
  **return** $T$, $P$;
**End Function**

---

## Appendix C  Basic statistics of the inferred molecular NCE grammars

First, we report the basic statistics of the molecular NCE grammars inferred from the ZINC250k dataset.

To check the generalization ability of the molecular NCE grammars, we parsed the molecules in the training data. From the 220,011 training molecules, we obtained 1,775 production rules. To investigate the coverage rate of the grammar, we parsed the 5,000 molecules in the test data using the production rules inferred from the training data to estimate the percentage of molecules that cannot be represented by the inferred grammar. The result shows that only 3 out of the 5,000 molecules cannot be parsed. Our coverage rate is higher than the one achieved by the MHGs and the number of our production rules is less.

Next, we inferred grammatical production rules from all 250k ZINC250k molecules, resulting in 1,838 production rules in total. Each molecule is associated with 28 production rules on the average. The maximum number of production rules associated with a molecule is 51.

For the antibiotic dataset, we extracted production rules from all known molecules. We parsed the molecules starting from different nodes to extend the number of training production sequences. 3,897 production rules were obtained from the dataset.

For the data provided by GuacaMol, 7256 production rules were obtained from the training set, leading to 293/238708 molecules in the test set uncovered. In comparison, 13110 production rules were obtained for the MHGs and 1088/238708 molecules were not covered by the inferred MHG.

When training the generation model, we set $L_{max}$ as the maximum number of production rules that a molecule in the dataset may be associated with. During the test, we set $L_{max}$ as $\infty$ in the experiments on ZINC250k and GuacaMol, but in the antibacterial experiment, considering the application scope of the classifier, we set $L_{max}$ the same as in the training process.

## Appendix D  Experimental settings of the baseline methods

Four state-of-the-art methods are compared with our method. 1) Junction tree VAE (JT-VAE) is a state-of-the-art algorithm for generating molecular graphs under the VAE framework. The basic idea of JT-VAE is to generate molecular graphs cluster by cluster and join each generated cluster using a greedy search. JT-VAE can generate molecules with 100% validity and it outperformed the previous methods such as Syntax-directed VAE and grammar VAE in property optimization and constrained property optimization. 2) Graph convolutional policy network (GCPN) aims to generate molecules atom by atom and optimize the properties of molecules by RL. As chemical validity cannot be guaranteed intrinsically in GCPN, it checks the validity of the graph in each step and discards invalid parts. A beam search is used in GCPN to improve sampling efficiency. GCPN achieved much better performance in property optimization, property range targeting and constrained optimization than the previous methods including JT-VAE. 3) Molecular hypergraph grammar variational autoencoder (MHG-VAE) uses molecular hypergraph grammars (MHGs) to assist the generation of molecular graphs and focuses on generating molecules with limited property evaluations. MHG-VAE uses an MHG as the prior of its VAE model, and achieved better performance than GCPN and JT-VAE under the limited property evaluation setting, but showed no advantage over other methods when property evaluation was unlimited. 4) Molecule Swarm Optimization (MSO) is a state-of-the-art algorithm in multi-objective molecular optimization with the particle swarm optimization algorithm and achieved excellent performance on the benchmarks provided by GuacaMol. The codes of the baselines were downloaded from GCPN, JT-VAE, MHG-VAE and MSO.

**Property optimization with unlimited property evaluations.** The results of GCPN were copied from [5]. As JT-VAE provided the molecules it generated when optimizing the penalized logP, we obtained the results directly by scoring the provided molecules. As for the task of optimizing QED, we set the objective function as the QED score and ran the code of JT-VAE with the default setting ten times to generate novel molecules. The results were obtained by summarizing all the molecules generated in the ten runs. For MHG-VAE, we copied its results in optimizing penalized logP from [3] and obtained the results in optimizing QED by running its code in the default setting with the QED score as the objective function. For MSO, to fairly compare with our method, the results in Table 1 were obtained by constraining the maximum number of atoms to 51 and the best hyperparameters

73  used in the corresponding paper [4] were adopted in our experiments. We ran MSO 100 times
74  and merge all the obtained molecules as the results. As a comparison, the results of MSO without
75  constraints on the number of atoms as well the results of our method under relaxed constraints are
76  shown in Table G.1.

77  **Constrained property optimization.** The results of all baselines were copied from the correspond-
78  ing papers [5, 3, 2].

79  **Comprehensive evaluations with GuacaMol.** The results of all baselines were copied from the
80  corresponding papers [4, 1].

81  **Property range targeting.** The results of GCPN and JT-VAE were directly copied from [5].

82  **Property optimization with limited evaluations:** The results of JT-VAE and GCPN were copied
83  from [3]. For MHG-VAE, we ran the code ten times and took the first 250 molecules each time, with
84  the same hyperparameters used in [3]. For MSO, we ran their code ten times and took the first 500
85  molecules each time, with the default hyperparameters.

86  # Appendix E    Evaluation of antibacterial properties

87  Enzymes are biological catalysts. A protease is an enzyme that performs proteolysis, that is, it triggers
88  protein catabolism by hydrolysis of the peptide bonds that link amino acids together in a polypeptide
89  chain. A kinase is an enzyme that catalyzes the transfer of phosphate groups from high-energy,
90  phosphate-donating molecules to specific substrates. As enzymes play an important role in bacterial
91  activities, molecules with high enzyme inhibitor scores, protease inhibitor scores or kinase inhibitor
92  scores are thought to be high-potential candidates for antibiotics.

93  The inhibitor scores were computed by using the Molinspiration online server. The larger the score
94  is, the higher is the probability that the involved molecule will be active. In particular, molecules
95  with positive scores are usually thought to be active. In our experiment, we adopted the thresholds
96  used by Molinspiration and regarded those with scores larger than 0.2 as active molecules and those
97  with scores larger than 0.5 as highly active molecules.

# Appendix F    Supplementary details of model training

The model is pre-trained with known molecules by maximizing the likelihood and then trained for each optimization task. The hyperparameters in reward functions are optimized for each task independently. For tasks with unlimited property evaluations, the other hyperparameters are optimized on the optimizing penalized logP task. The hyperparameters for the optimization with limited property evaluations are optimized independently. For each task, the best molecules found by the policy are used as known trajectories to train the model to accelerate convergence. With 1080Ti, the pre-training on ZINK250 took around 27 hours and the optimization stages took 30 minutes $\sim$ 24 hours depending on the tasks.

 **Appendix G   Supplementary results**

Figure G.1: The 20 molecules with the highest penalized logP scores generated by MNCE-RL in optimizing the penalized logP score with unlimited property evaluations. The diversity of 5000 molecules is 0.722.

Table G.1: The maximum penalized logP scores with different $L_{max}$ values. The results of MSO are copied from the corresponding paper [4].

| Method | Penalized logP | | | | | Validity |
|---|---|---|---|---|---|---|
| | $1^{st}$ | $2^{nd}$ | $3^{rd}$ | $50^{th}$ | Top 50 Avg. | |
| MSO (no constraints on the number of atoms) | 26.10 | - | - | - | - | - |
| MNCE-RL ($L_{max} = 51$) | 18.33 | 18.18 | 18.16 | 17.52 | 17.76 | **100%** |
| MNCE-RL ($L_{max} = 90$) | 28.09 | 28.04 | 28.00 | 26.52 | 26.99 | **100%** |
| MNCE-RL ($L_{max} = 110$) | **34.06** | **34.04** | **33.92** | **32.96** | **33.33** | **100%** |

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

QED=0.948    QED=0.948    QED=0.948    QED=0.948

QED=0.948    QED=0.948    QED=0.948    QED=0.948

QED=0.948    QED=0.948    QED=0.948    QED=0.948

QED=0.948    QED=0.948    QED=0.948    QED=0.948

QED=0.948    QED=0.948    QED=0.948    QED=0.948

Figure G.2: The 20 molecules with the highest QED scores generated by MNCE-RL in optimizing the QED score with unlimited property evaluations. The diversity of 5000 molecules is 0.870.

Target, penalized logp=-1.09    Similarity=0.44, penalized logp=3.09    Similarity=0.67, penalized logp=1.48

Target, penalized logp=-3.24    Similarity=0.44, penalized logp=0.33    Similarity=0.66, penalized logp=-1.55

Target, penalized logp=-4.27    Similarity=0.41, penalized logp=0.68    Similarity=0.6, penalized logp=-1.03

Target, penalized logp=-1.61    Similarity=0.41, penalized logp=2.28    Similarity=0.61, penalized logp=0.16

Target, penalized logp=-2.35    Similarity=0.4, penalized logp=3.27    Similarity=0.61, penalized logp=1.93

Figure G.3: Five target molecules (the first column) in constrained optimization and their corresponding optimized molecules generated by MNCE-RL (the second and the third columns).

Figure G.4: The best penalized logP scores of the molecules found by different methods depending on the number of function evaluations.

Penalized logp=9.88          Penalized logp=9.82          Penalized logp=9.75          Penalized logp=9.67

Penalized logp=9.66          Penalized logp=9.65          Penalized logp=9.43          Penalized logp=9.4

Penalized logp=9.21          Penalized logp=9.1           Penalized logp=8.85          Penalized logp=8.84

Penalized logp=8.73          Penalized logp=8.68          Penalized logp=8.68          Penalized logp=8.6

Penalized logp=8.59          Penalized logp=8.59          Penalized logp=8.43          Penalized logp=8.33

Figure G.5: The 20 molecules with the highest penalized logP scores generated by MNCE-RL in optimizing the penalized logP score with limited property evaluations.

KI = -0.35, PI = 0.39, EI = 0.06    KI = -0.27, PI = 0.41, EI = 0.10    KI = -0.38, PI = 0.06, EI = 0.32

KI = -0.18, PI = -0.10, EI = 0.49    KI = -0.24, PI = 0.66, EI = 0.30    KI = -0.16, PI = 0.66, EI = 0.16

KI = -0.24, PI = 0.63, EI = 0.09    KI = -0.34, PI = 0.55, EI = 0.15    KI = -0.20, PI = 0.54, EI = 0.23

KI = -0.38, PI = 0.56, EI = 0.25

Figure G.6: The ten molecules with the highest scores assigned by the classifier in generating candidates of antibiotics and their corresponding property scores.