[Reviews · NeurIPS 2020]

Review 1

Summary and Contributions: Guiding the design and optimization of molecules with desired properties is a recent application area of machine learning that builds on recent success in predicting such properties. The authors propose a method that uses reinforcement learning with a graph convolutional network as its policy network to explore molecule space using actions suggested by a graph grammar. They obtain state-of-the-art results on several benchmark datasets.

Strengths: + The authors combine ideas from several recent publications addressing this problem - the use of reinforcement learning in conjunction with GCNs (ref [34]) and graph grammars (ref [16]). + The results suggest a strong improvement over recently published work. + Code is provided with the submission that appears ready for upload to a repository such as github.

Weaknesses: - The authors claim that a strength of their method is limited number of property evaluations. It's not clear why that is important, as that is performed computationally.

Correctness: As far as i can tell.

Clarity: Yes.

Relation to Prior Work: Yes.

Reproducibility: Yes

Additional Feedback:


Review 2

Summary and Contributions: The present paper is concerned about molecular optimization and proposes the method called MNCE-RL, a reinforcement learning agent assisted by the neighborhood-controlled embedding (NCE) grammar tailored for molecules. The existing NCE grammar cannot deal with molecular graphs mainly because it does not support edge labeling and the valency conditions may be violated. The authors propose a novel graph grammar called the molecular NCE grammar so as to address the aforementioned issues. The authors demonstrate the effectiveness of the proposed method in common benchmark tasks, and empirically validate that MNCE-RL achieves better performance than baseline methods.

Strengths: (S1) The authors employ four common benchmark tasks and one realistic benchmark task to empirically show the performance gain of the proposed method. Since the proposed method consistently outperforms the baseline methods, these results will be a strong evidence indicating the superiority of the proposed method over the methods compared. (S2) This paper is very relevant to the NeurIPS community in that there are tens of papers in the machine learning community solving the same problem.

Weaknesses: (W1) The empirical evaluation is not much convincing because one previous paper entitled "Efficient multi-objective molecular optimization in a continuous latent space" reports better score than the proposed method. In specific, the best penalized logP score reported in the paper is 26.1, while that of the present paper is 15.18. If the authors are to claim that the proposed method achieves the state-of-the-art score, the paper mentioned above should be mentioned and compared empirically. (W2) The authors provide no rationale behind the experimental results. Without any insight about why the proposed method is better than the existing methods, readers are difficult to learn lessons from the paper, which severely limits the impact of the paper; if there was such an insight, readers would be able to apply the insight to other domains appropriately so as to obtain performance gain. For example, the reviewer is curious about (i) how much we benefit from the proposed grammar as compared to the molecular hypergraph grammar and why it is, (ii) which of the RL module or the grammar of the proposed method contributes to performance improvement, (iii) why the proposed method is more sample-efficient than the baseline methods, etc. If the authors provided insights into these questions, they would be much beneficial to further enhance the performance of molecular optimization methods.

Correctness: The empirical methodology needs improvement. One concern is the validity of the experiment setting dealing with the antibacterial property. Instead of wet-lab experiments, the authors use a classifier trained on a real-world data set as a pseudo evaluation function, and the authors state that "the classifier is severely overfitted". Although the authors tried to alleviate the overfitting issue, no evidence seems to be provided to validate the use of the pseudo evaluation function. Another suggestion is to use a common benchmark environment such as GuacaMol. One issue in the molecular optimization research is that each paper reports experimental results on a little bit different experimental setting, limiting us to compare different methods on the exactly same tasks. To alleviate this, several research groups propose benchmark task suites, and the reviewer recommends the authors to employ some of them.

Clarity: The reviewer feels it difficult to understand the technical details of the present paper. i) The relationship between the existing NCE grammar and the proposed one is not clear. One clear difference is that every node in the LHS of a production rule of the existing one is removed, while some of the nodes in the LHS of a production rule of the proposed one, Bp, are not removed, because they serve as "neighbors" in the original grammar. Given this difference, the reviewer had difficulty in understanding the relationship. Is the proposed one a special case of the existing one or the other way around? ii) Although the authors claim that "the valency validity can be guaranteed intrinsically", the reviewer had difficulty in confirming this. The reviewer wishes to see a mathematical proof on this as well as its intuition.

Relation to Prior Work: As discussed in the weaknesses section, this work partially lacks the relationship to the literature. In order to claim that the proposed method achieves the state-of-the-art performance, the paper mentioned above should be contrasted with the proposed method.

Reproducibility: Yes

Additional Feedback: I appreciate the authors for addressing most of my concerns. I have updated my score from 4 to 6. i) For the empirical evaluation, I understand that the proposed method performs better than the method I found, when compared in fair settings. ii) For the insight of performance improvement, I partially agree with the authors' insight, but it is still unclear without the results of the ablation study. iii) For the experiment on the antibacterial property, this was completely my misunderstanding. I think the experimental setting is sound enough, because the evaluation score is independent of the classifier. iv) For the common benchmark environment, I understand that it is very time consuming to run the GuacaMol benchmark. I wish the authors mention the existence of such benchmark environments in the main text so that following papers can use them. v) For the graph grammar, I understand that valency is preserved when the inferred grammar is used. I would like the authors to clarify that the valency-preserving property comes from the inference algorithm rather than the definition of the molecular NCE grammar, because Definition 1 does not much specify the embedding function phi. For example, if we add phi(1, 6)="..." in the production rule shown in the top of Figure 2, this production rule does not preserve the degree of node 1, while the embedding function with phi(1, 6)="..." is still legal. vi) I would like the authors to improve explanations of the proposed grammar and inference algorithms. For example, while I could roughly understand the grammar and the inference algorithms and I could partially reproduce production rules shown in Figure A.2 (where l.h.s. of rule (7) seems not to be correct because there is no "x"), I could not understand the followings: (a) why n_Sigma appears in the rhs of a production rule. As far as I understand, n_Sigma in lhs of a production rule is necessary but n_Sigma in rhs of a production rule should be replaced with x (in other words, one non-terminal node label, x, is enough in rhs of a production rule and intermediate graphs, and n_Sigma is only necessary in lhs of a production rule). This question comes from Figure A.3, where x in lhs of a production rule matches n_Sigma in an intermediate graph when applying rule (4). (b) how v_t (the next non-terminal to be replaced) is determined during graph generation and inference algorithms, (c) the relationship between the proposed grammar/inference algorithm and the molecular hypergraph grammar [16]. I am especially curious about (c); one obvious difference is that MHG does not divide ring structures while the proposed algorithm does. However, they have similar principles that they try to obtain a skeleton of a molecule. In addition, production rules of the proposed grammar seem to be compatible with production rules of a hyperedge replacement grammar. This can be intuitively confirmed by regarding node x as a non-terminal hyperedge and node n_Sigma as nodes involved in the hyperedge. Given this correspondence, the proposed grammar seems to be a bit less innovative than I thought at first sight (still I admire the technical achievement of this work). Therefore, I would like the authors to discuss the relationship between the proposed grammar and MHG.


Review 3

Summary and Contributions: This work describes the use of a Neighborhood-Controlled Graph Grammar for molecular design. The approach is based on a canonical parse-tree decomposition of the graph that is expressed then as actions or "production rules", that act on intermediate graphs and update them. This sequential graph construction approach lends itself to reinforcement-learning approaches where the intermediate graph is updated to match a certain target. The key contributions of the paper are transferring the idea of NCE grammar from graphs in general to molecular graphs and addressing some unique challenges. Some benchmarks are also shown for the approach.

Strengths: This is a very original approach. The world of NCEs and molecular design have not been connected before, to my knowledge. This will further intertwine the graph representation learning / graph generation and molecular design communities, who have typically found good success in sequential models, but acting on string representations. The mathematical descriptions of the approach are thorough. Some experiments are provided. In the Broader impact section: "guaranteed to possess desirable biological or chemical properties" What ist meant by guarantees?

Weaknesses: There are hundreds if not thousands of generative / optimization models for molecules, and it is vital for the field to compare against common benchmarks. Here, the metrics are reported against very simple, weak metrics (logP) and on a number of new tasks ("we designed a series of experiments a"). It would be much more beneficial to utilize the metrics and performance baselines from either Guacamol (10.1021/acs.jcim.8b00839) or MOSES (arxiv.org/abs/1811.12823). If every work comes with their own metric, then it is nearly impossible to have true comparison, particularly when the proposed tasks are artificial and not related to a realistic chemical optimization problem (the molecular weight task, for instance) One of the strengths of generative algorithms for molecules is to capture a hard-to-describe statistical distribution of plausible molecules that can be made, afforded, stored, etc. Reinforcement learning takes out the rails and is likely to generate molecules well out of sample for the training distribution. It is of extra importance for these approaches to describe whether the generated molecules “make sense” and why or why not. For instance, far-away chemical moieties in the molecule may be incompatible because the would react with each other, so a global state/action context is needed. Many of the reported molecules in the SI are not correct, and are only “valid” because of bugs in RDKit. The limited-evaluations issue is not really a limiting factor in the area, where property prediction is typically based on some learnable or proxy model that can be evaluated millions of times. Why is there no diversity metric in the QED task? The molecules in the SI all look extremely similar. Kekulé is mistyped. Edit: The authors have provided feedback where they commit to addressing most of the issues I raised above.

Correctness: The Diversity is the average distance or the average similarity? No molecules are reported in the SI (they are buried in the attached code, it is appreciated that they are given). Mean distance of 0.92 in Tanimoto distance over fingerprints is uncommon and would mean molecules that hardly share any bits. There may be some issues. Extremely different molecules typically have 0.5 distance. What radius, what length fingerprint. Is it possible to show such molecules in the SI and discuss? The molecules look extremely similar. Edit: This was addressed in the author feedback. This is an Achilles heel of this paper: Node labels are not independent of edge labels in molecules. Aromaticity for instance is not a fully resolved issue and RDKit resorts to some conventions that break down at inference time. The proposed approach struggles to annotate bond orders and aromaticity correctly. It is unfortunate [collaboration with domain experts would have helped prevent this] but the example molecule in the SI is not a valid chemical graph, even if rdkit might fail to detect it, the number of electron pairs in each atoms is just not correct: 5-valence carbon. This hints at very deep underlying issues in the model, if it has not been able to learn that carbon is at top 5-connected it is hard to expect that more complex rules would be captured. This molecule, for instance (from the 525 Mw experiment) is completely non-sensical too. CCC(=O)[C@H](NC)[C@H](C)N[C@H](C)C(C)=CC(C)=CC(=NC(NC)=S1(=S)C(C)=N[C@@H]1CNN(C)C)NC The model is learning hard-coded rules from rdkit but not capturing any higher-level complexity.

Clarity: The paper is overall clear and well written, but it would have been helpful to have a more high level description of the approach for the broader audience.

Relation to Prior Work: Yes

Reproducibility: Yes

Additional Feedback: The second point raised in the Correcteess section (Node vs. edge labels) has not been addressed. I encourage the authors to at least describe it and propose future work that could address it.


Review 4

Summary and Contributions: The authors MNCE-RL, a method for generating molecules with desired properties that extends the neighborhood controlled embedding grammar to i) take edge labels into account, ii) taking connections between subgraphs into account, and iii) decreasing the number of production rules. A molecule is generated sequentially by a policy trained via PPO conditioned on the graph generated so far, which is encoded by a graph convolutional network. MNCE-RL guarantees generating valid molecules and outperforms baselines in optimizing properties of molecules.

Strengths: The paper is clearly written, has a clear methodological contribution, and experimental results are promising.

Weaknesses: 1. What is the runtime of MNCE-RL compared to existing approaches based on graph CNNs? 2. PPO is an on-policy approach, which requires that data used to update the policy are collected by the same policy. Can MNCE-RL use initial (graph, reward) pairs that were collected be a different policy? Labeled data are available in many practical optimization settings but it is unclear if MNCE-RL can make use of these data. 3. l222: 'We report the top 3 property scores, the 50th best score, and the average score of the top 50 molecules'. These metrics can be optimized by generating repeatedly the same molecule with a high score or very similar molecules. However, desired is the ability to find many diverse molecules with a high score. Do you take a minimum distance between generated molecules into account when computing these metrics? 4. Section 'Property optimization with limited property evaluations'. Instead of reporting the best scores after a fixed number (500) of function evaluations, I suggest plotting the maximum of f depending on the number of function evaluations. This will show more clearly how fast methods optimize f and does not require choosing a certain number of problem evaluations. 5. Section 'Generation of novel molecules with antibacterial property'. It is unclear if the classifier that is optimized by MNCE-RL is a the ground truth f(m), or a surrogate f'(m) for the unknown and costly to evaluate objective function f(m) (antimicrobial activity quantified experimentally). In. the first case, it is necessary to report the number of function evaluation performed and comparing MNCE-RL for making the claim 'illustrates the ability of MNCE-RL to generate antibacterial candidate molecules with only limited labeled samples'. In the latter case, the number of function evaluation is negligible since the function is inexpensive to evaluate (it is a model) and can be also optimized by other methods (e.g. JT-VAE). 6. How and which hyper-parameters of MNCE-RL and baseline methods were optimized?

Correctness: The methodology seems correct although I am not an expert in this field

Clarity: The paper is clearly written.

Relation to Prior Work: Related work is discussed though I can not judge if it is comprehensive since I am not an expert in this field.

Reproducibility: Yes

Additional Feedback: 7. L25: 'The second challenge (sample-efficiency) is addressed by BO and RL'. I agree that BO is sample-efficient but RL is usually considered as sample-inefficient. 8. L26: 'Few of these methods consider the the high cost to evaluate molecular properties'. This contradicts the previous sentence. BO is widely used for optimizing functions that are costly to evaluate. 9. L104-108: How sensitive is the performance to the choice of r_\eps?

[Author Response · NeurIPS 2020]

The following is our response to all major comments.

**Baselines and benchmarks** (Reviewers #2 and #3): We appreciate the reviewers' input on additional baselines and benchmarks. The reason that the method MSO in "Efficient multi-objective molecular optimization in a continuous latent space" achieved a higher penalized logP with unlimited property evaluations than ours (26.1 vs 15.18) is due to different experimental settings. The best achievable penalized logP score in this experiment is highly correlated with the maximum number of atoms allowed, which, in our method, is determined by the maximum number of steps ($L_{max}$). We set $L_{max} = 51$ in the experiment to be consistent with the training data (Appendix C, as commonly done in the literature), while the top molecules presented in the above paper involved many more atoms than 50. With a larger $L_{max}$, the best penalized logP score can be significantly increased. For instance, with $L_{max} = 110$, our top penalized logP score increases to 32.38 and by limiting MSO to molecules of at most 50 atoms, its top penalized logP score decreases to 14.19. We will include a fair comparison with the new baseline in the final version. We have started running the experiments on GuacaMol as suggested. But, due to the size of the data and number of objectives, we estimate that the full experiments will take several weeks to complete and hope to report the results in the final version.

**Insight into the performance improvements** (Reviewer #2): Compared with a molecular hypergraph grammar (MHG), our proposed method has a higher coverage rate (in Appendix C, MHG failed to cover 16 of the 5000 molecules while our method only failed to cover two). Thus, our inferred grammar can represent more molecular structures and explore the chemical space more effectively. Compared with atom-by-atom methods such as GCPN, our method is much more sample-efficient since the validity of the generated structure is regularized by the grammar and the deep-learning model can focus on property optimization. Moreover, we think that the ability to update bond features in the GCN also helps to improve the performance. These insights as well as a suitable ablation study will be added in the final version.

**Validity and evaluations of generated molecules** (Reviewers #3 and #4): We apologize for overlooking the validity of the molecules presented in Figures A.2 and A.3 containing a valence-5 carbon. This molecule is only intended for clarifying the inference and derivation steps of our proposed grammar and is **not** among the molecules generated by our method. We will fix these two figures in the final version. All generated molecules in the appendix have been double-checked by both RDkit and human experts. Since almost all previous publications used RDkit to check the validity of generated molecules, we think it is a reasonable measure. On the other hand, we agree that some unstable structures might be missed by RDkit, which is a well-known problem in molecular optimization with ML methods.

The diversity of the generated molecules in optimizing QED and penalized logP with unlimited property evaluations are both 0.8, and we will add these in the final version. Following ORGAN and GCPN, the diversity within generated molecules is calculated as the average pairwise Tanimoto distance between the Morgan fingerprints of the molecules. Same as previous work, the radius is 4, the number of bits is 2048, and the range of the obtained diversities is consistent with the values presented in the paper of GCPN. Thus, we believe our results are reliable.

**Limited number of property evaluations** (Reviewers #1, #3 and #4): We highlight that our method can optimize molecules with a limited number of property evaluations because in many real-world biological and biomedical applications, the required property evaluations can be very expensive while efficient and accurate proxy models are unavailable. For instance, in the development of novel drugs, wet-lab experiments or time-consuming computations are usually needed to evaluate the properties of a molecule, and thus a model's ability to perform optimization with a limited number of property evaluations is crucial and can significantly reduce the cost. But, we appreciate the Reviewer 4's suggestion of plotting the results of this experiment and plan to add a plot in the final version.

**Antibacterial experiment** (Reviewers #2 and #4): This experiment is meant to illustrate an application of our method to a real-world problem. A trained classifier is used as a surrogate for the unknown property evaluation function as wet-lab experiments were not available in our lab. Since the evaluation method (inhibitor scores) of generated molecules is independent from the pseudo evaluation function, we think that the molecules generated by our method with high inhibitor scores can still be used as candidate antibacterial molecules.

**Relationship with the original NCE grammars** (Reviewer #2): We have made significant modifications to the original NCE grammars especially concerning the embedding function and LHS of a production. It is easy to see that the valency validity cannot be guaranteed by the original NCE grammars. In our proposed grammars, we constrain the form of an LHS to a subgraph consisting of only *a node and its neighbors* to simplify the production rules and our embedding function defines the labeled connections between these neighbors and the nodes in the RHS. As the bonds connected to an atom remain constant throughout molecule generation process and the production rules are chosen based on known molecules, the valency validity can be guaranteed.

**Model training details and running time** (Reviewer #4): Since the classical PPO is used in our method, off-policy training is not supported. On the ZINC dataset, the pre-training of our method took less than 24 hours and each optimization task took less than 12 hours. More details of running time comparison as well as hyper-parameter optimization will be included in the final version.

[Meta-Review · NeurIPS 2020]

The reviewers find a compelling new method with good empirical results.